# Source-Optimal Training is Transfer-Suboptimal

**C. Evans Hedges** *evans@hedgesfamily.com*
*Independent Researcher*

**Reviewed on OpenReview:** *CMlpokFXfA*

## Abstract

We prove that training a source model optimally for its own task is generically suboptimal when the objective is downstream transfer. We study the source-side optimization problem in L2-SP (L2-distance to Starting Point) ridge regression, where the target estimator is regularized toward the source model parameters, and show a fundamental mismatch between the source-optimal regularization $\tau_S^*$ (minimizing source risk) and the transfer-optimal regularization $\tau_0^*$ (maximizing downstream transfer): outside of a measure-zero set, $\tau_0^* \neq \tau_S^*$. We characterize $\tau_0^*$ as a function of the normalized task alignment $\rho = \langle w_0, w_1 \rangle / \|w_0\|^2$ and identify an alignment-dependent reversal: with imperfect alignment ($0 < \rho < 1$), transfer benefits from stronger source regularization, while in super-aligned regimes ($\rho > 1$), transfer benefits from weaker regularization. In isotropic settings, whether transfer helps is independent of target sample size and noise. We verify the phase transition in synthetic experiments across overparameterization ratios and covariance structures, and present nonlinear experiments on MNIST, CIFAR-10, and 20 Newsgroups showing that the mismatch persists in standard transfer learning pipelines, with explicit L2-SP fine-tuning closely tracking standard SGD and the target sample-size independence prediction confirmed empirically.

## 1 Introduction

### 1.1 Background and Motivation

A fundamental question in transfer learning is source-side optimization: how should we train the source model if the objective is downstream transfer? We study this question in the L2-SP (L2-distance to Starting Point) ridge regression framework, where the target model is regularized toward the source model parameters (we define the setup precisely in Section 2). We prove a source-optimal versus transfer-optimal mismatch: the source regularization $\tau_S^*$ that is optimal for the source task is generically suboptimal for transfer, and outside the set $\{\rho = 1\}$ we have $\tau_0^* \neq \tau_S^*$, where $\tau_0^*$ denotes the transfer-optimal source penalty and $\rho = \langle w_0, w_1 \rangle / \|w_0\|^2$ is the normalized alignment between the source and target task vectors $w_0, w_1$. This mismatch is alignment-dependent: in the standard imperfect-alignment regime ($0 < \rho < 1$), transfer benefits from stronger source regularization than source-optimal training, while in a super-aligned regime ($\rho > 1$) transfer benefits from weaker regularization.

Intuitively, this divergence arises from a geometric mismatch between the source signal and the useful signal for transfer. The source estimator $\hat{\beta}_0$ naturally targets $w_0$, but the optimal anchor for the L2-SP penalty on the target task (the best fixed point toward which to regularize the target estimator) is the projection $\rho w_0$. In the imperfect-alignment regime ($0 < \rho < 1$), the source estimator is systematically "too large" relative to this projection; stronger regularization is required to shrink it towards the optimal scale. Conversely, in the super-aligned regime ($\rho > 1$), the source estimator is "too small" and weaker regularization is preferred to preserve more signal magnitude, even at the cost of higher variance. This scaling requirement exists independently of observation noise, though noise further modulates the optimal set-point. Our claims are exact for linear ridge regression; our contribution is isolating a source-side mechanism and making it analytically sharp.

We begin our exploration through the lens of evaluating when we would expect positive or negative transfer from some source task to a target task. Previous work Dar & Baraniuk (2022); Lampinen & Ganguli (2018); Yang et al. (2025) has shown that the phenomenon of negative transfer is a concern, where the model generated by transfer learning performs worse than if one were to train the model from scratch. Intuition and previous theoretical work Lampinen & Ganguli (2018) suggest that if the tasks and input datasets are similar enough, transfer learning techniques should help, but if the target task is sufficiently different, or in the context of continual learning the regime has shifted enough, transfer learning may harm the final model, introducing bias and noise from the initialization.

There has been notable work examining this phase boundary between when one should expect positive or negative transfer. In Dar & Baraniuk (2022); Dar et al. (2024), Dar and Baraniuk provided an explicit quantitative phase boundary for freezing/parameter-sharing style transfer in linear models, giving if-and-only-if conditions for when shared representations outperform training from scratch. In Dhifallah & Lu (2021), Dhifallah and Lu explored the phenomenon in the context of feature freezing. Additionally, feature-overlap driven transitions in linear transfer were explored by Tahir, Ganguli, and Rotskoff in Tahir et al. (2024). Related phase transitions have been characterized for other linear transfer mechanisms such as hard parameter sharing Yang et al. (2025). Complementary asymptotic analyses for fine-tuning from pretrained anchors via gradient descent appear in Ghane, Akhtiamov, and Hassibi Ghane et al. (2024), who provide universal asymptotics comparing pretrained versus fine-tuned models under distributional shifts.

The L2-SP approach was introduced by Li, Grandvalet, and Davoine Xuhong et al. (2018), who showed empirical improvements by penalizing deviation from pre-trained parameters. This approach matches many practical fine-tuning protocols and corresponds to the isotropic elastic weight consolidation (EWC) penalty in Kirkpatrick et al. (2017). Dar, LeJeune, and Baraniuk Dar et al. (2024) recently analyzed when L2-SP transfer learning outperforms standard ridge regression, assuming task parameters are related by orthonormal transformations and focusing on optimal tuning of the target-task regularization given a fixed source model.

Our work differs from these prior analyses by solving the source-side optimization problem, revealing that the transfer-optimal source regularization $\tau_0^*$ generically differs from the source-task-optimal choice. We additionally verify empirically that this phenomenon persists across a range of nonlinear networks and tasks.

## 1.2 Setup and Overview of Results

In this paper we formalize transfer learning from the lens of L2-SP ridge regression. In particular, we have two tasks, Task 0 (the source task) and Task 1 (the target task), as well as corresponding training datasets $(X_0, y_0)$ and $(X_1, y_1)$. We then seek to evaluate the expected out-of-sample risk on Task 1, comparing the ridge/ridgeless solution trained solely on $(X_1, y_1)$ to the L2-SP ridge solution found by first training a ridge/ridgeless model on $(X_0, y_0)$, and then using those model parameters as the prior for ridge/ridgeless training on $(X_1, y_1)$.

Our analysis builds on the foundational work characterizing ridge and ridgeless regression risk in high-dimensional settings Dobriban & Wager (2018); Hastie et al. (2022), which established the precise asymptotic behavior of these estimators in overparameterized regimes and revealed phenomena such as double descent Belkin et al. (2019) and benign overfitting Bartlett et al. (2020). We employ random matrix theory techniques, specifically the deterministic equivalent framework Bai & Silverstein (2010); Couillet & Debbah (2011); Dobriban & Wager (2018); Hastie et al. (2022), to derive precise asymptotic characterizations of the estimators' risk and identify sharp phase boundaries for transfer benefit.

Our contributions can be outlined as follows. First, Theorem 3.3 provides an if-and-only-if inequality describing when transfer learning will outperform from-scratch training in the finite data, non-isotropic ridge regime. As a corollary, we show that in the finite sample, isotropic, ridgeless case the inequality takes a simple form and is interestingly independent of $n_1$ and $\sigma_1$. This independence is consistent with prior isotropic analyses of freezing-style transfer Dar & Baraniuk (2022); Lampinen & Ganguli (2018), and our finite-sample isotropic/ridgeless corollary recovers these patterns in the L2-SP setting.

We then examine the asymptotic limit using deterministic equivalents (DEs), a standard tool from random matrix theory for characterizing the limiting behavior of random quadratic forms. Theorem 3.6 provides

a DE characterization and asymptotic boundary for L2-SP ridge transfer with general (non-orthonormal) task vectors, where the overparameterization ratio $\gamma_i = p/n_i$ and noise level $\sigma_i$ govern the risk. Corollary 3.7 examines the isotropic case, where the decision criterion is determined entirely by whether or not the alignment of the two tasks surpasses a bias and noise term dependent only on Task 0.

From this we arrive at Theorem 3.8, which identifies the unique $\tau_0^*$ (source model ridge penalty) that maximizes transfer benefit, given a particular task alignment. Outside of a measure-zero set, $\tau_0^*$ does not coincide with the optimal Task 0 ridge parameter, and notably $\tau_0^*$ is independent of target sample size but depends on task alignment. Together, these results give the first explicit analytical boundary for when Euclidean L2-SP transfer helps with general (non-orthonormal) task vectors in overparameterized ridge models, including general covariance and DE limits, and provide surprising insights into optimally training source models for the purpose of transfer learning.

Finally, we establish in Corollary 3.9 an alignment-dependent phase transition for the optimal source penalty. Under standard imperfect task alignment conditions, $\tau_0^*$ is always strictly greater than the source-optimal ridge penalty. However, in super-aligned regimes this relationship reverses. This implies that maximizing transfer benefit requires adjusting regularization based on the geometric relationship between tasks, rather than optimizing for source performance alone.

Though derived in linear models, these results isolate a core mechanism that may persist in overparameterized nonlinear networks. Additionally, although standard SGD fine-tuning does not use an explicit L2-SP penalty, the optimization trajectory remains anchored in the basin of the pretrained parameters, functionally approximating the L2-SP constraint. In Section 4 we validate the phase transition in synthetic ridge experiments and then probe source-optimal versus transfer-optimal regularization in nonlinear networks on MNIST, CIFAR-10, and 20 Newsgroups, finding consistent over-regularization for transfer across all tested domains.

These results challenge the conventional practice of optimizing source models solely for their own performance. For practitioners training foundation models intended for transfer, our analysis suggests that regularization strategies should explicitly account for the downstream transfer objective and source data quality.

The remainder of the paper is outlined as follows. Section 2 sets up the model and assumptions. Section 3 presents the core mathematical results. Section 4 provides empirical validation of our theoretical predictions on synthetic data, MNIST, CIFAR-10, and 20 Newsgroups. We conclude with Section 5 and discuss implications and limitations; proofs are deferred to the Appendix.

## 2   Preliminaries

We let $Z_0, Z_1$ be $n_0, n_1 \times p$ random matrices with entries taken iid with mean 0, variance 1, and finite $2 + \epsilon$ moments and Lindeberg condition (see Bai & Silverstein (2010) for more information). These assumptions are sufficient for the deterministic equivalents we will examine later, but it is safe also to simplify these assumptions taking entries in $Z_i$ to be iid $N(0,1)$. We additionally assume we are in the overparameterized regime, with $n_0, n_1 < p - 1$. We let $\Sigma_0, \Sigma_1$ be covariance matrices and $X_i = Z_i \Sigma_i^{1/2}$. Next we have true signal vectors $w_0, w_1$ and let $y_i = X_i w_i + \epsilon_i$ where $\epsilon_i \sim N(0, \sigma_i^2 I)$. We assume the signal norms $||w_0||, ||w_1||$ remain bounded (i.e., $O(1)$) as $p \to \infty$, ensuring that the signal and noise contributions to the risk remain comparable in the asymptotic limit.

We will frequently quantify task relatedness by the normalized alignment

$$\rho := \frac{\langle w_0, w_1 \rangle}{||w_0||^2}.$$

We refer to $0 < \rho < 1$ as the *imperfect alignment* regime and $\rho > 1$ as the *super-aligned* regime.

We adopt the L2-SP approach found in Xuhong et al. (2018) and examine the following estimators:

$$\hat{\beta}_0(\lambda_0) := \arg\min_\beta \ \|y_0 - X_0\beta\|^2 + \lambda_0\|\beta\|^2$$
$$= (X_0^\top X_0 + \lambda_0 I)^{-1} X_0^\top y_0,$$

$$\hat{\beta}_1^{\mathrm{S}}(\lambda_1) := \arg\min_\beta \ \|y_1 - X_1\beta\|^2 + \lambda_1\|\beta\|^2$$
$$= (X_1^\top X_1 + \lambda_1 I)^{-1} X_1^\top y_1,$$

$$\hat{\beta}_1^{\mathrm{TL}}(\lambda_1 \mid \hat{\beta}_0) := \arg\min_\beta \ \|y_1 - X_1\beta\|^2 + \lambda_1\|\beta - \hat{\beta}_0(\lambda_0)\|^2$$
$$= (X_1^\top X_1 + \lambda_1 I)^{-1}\big(X_1^\top y_1 + \lambda_1 \hat{\beta}_0(\lambda_0)\big).$$

We also will take the notation that $\hat{\beta}_0(0) = \lim_{\lambda_0 \searrow 0} \hat{\beta}_0(\lambda_0)$ represents the ridgeless estimator (and similarly for $\hat{\beta}_1^S$ and $\hat{\beta}_1^{TL}$). In this setting $\hat{\beta}_0$ represents our ridge/ridgeless estimator for Task 0, $\hat{\beta}_1^{\mathrm{S}}$ is the standard (from-scratch) ridge/ridgeless estimator for Task 1 (the superscript S stands for "standard," not "source"), and $\hat{\beta}_1^{\mathrm{TL}}$ is our transfer learning estimator for Task 1 that takes the solution of Task 0 as its prior.

We note here that $\hat{\beta}_1^{\mathrm{TL}}(\lambda_1 \mid \hat{\beta}_0)$ corresponds exactly to the MAP estimator with Gaussian prior $\beta \sim N(\hat{\beta}_0, \lambda_1^{-1}I)$ and matches practical L2-SP fine-tuning Xuhong et al. (2018) and the isotropic EWC penalty Kirkpatrick et al. (2017). While our theoretical analysis assumes an explicit L2-SP penalty, standard fine-tuning protocols rely on the implicit regularization of SGD initialized at the pretrained parameters $\hat{\beta}_0$. For limited training horizons, the optimization trajectory remains anchored in the basin of $\hat{\beta}_0$, functionally approximating the L2-SP constraint. Thus, we expect the regularization-variance trade-offs identified in our ridge analysis to persist in standard fine-tuning. We also note that we do not reweight our penalty by a task metric $H$ and stick with standard Euclidean distancing for our ridge penalty. This differs from the whitened/metric-based formulations and also more accurately reflects typical implementations of L2-SP and fine-tuning where the penalty is applied in Euclidean parameter space without whitening.

To evaluate out-of-sample performance of these estimators we will use their expected prediction risk:

$$\mathrm{Risk}_1(\beta) := \mathbb{E}_{x_1 \sim N(0, \Sigma_1)}[(x_1^\top \beta - x_1^\top w_1)^2] = \|\beta - w_1\|_{\Sigma_1}^2,$$

where $\|v\|_\Sigma^2 = v^\top \Sigma v$.

Our first goal is to understand when L2-SP transfer improves expected Task 1 risk, namely when

$$\mathrm{Risk}_1(\hat{\beta}_1^{\mathrm{S}}(\lambda_1)) > \mathrm{Risk}_1(\hat{\beta}_1^{\mathrm{TL}}(\lambda_1 \mid \hat{\beta}_0(\lambda_0))).$$

We will additionally make use of the Frobenius norm in $\Sigma$ geometry, which we will denote as follows:

$$\|A\|_{\Sigma,F}^2 = \mathrm{Tr}\left(A^\top \Sigma A\right).$$

Finally, some common matrices we will be using deserve their own notation, and we define that here: let

$$M_{\lambda_1}^{(i)} = \left(X_i^\top X_i + \lambda_1 I\right)^{-1}$$

and note that

$$M_{\lambda_1}^{(1)} X_1^\top X_1 - I = -\lambda_1 M_{\lambda_1}^{(1)}.$$

Additionally let

$$P_i = X_i^+ X_i,$$

where $X_i^+$ is the Moore-Penrose pseudoinverse, so that $P_i$ is the orthogonal projector in $\mathbb{R}^p$ onto $\mathrm{row}(X_i)$ and $I - P_i$ projects onto $\ker(X_i)$.

# 3 Mathematical Results

All proofs are deferred to the Appendix; here we state each result and provide a brief proof sketch or key idea.

## 3.1 Finite Sample Risk Formulas

First we recall the expected out of sample risk of ridge regression (see Hastie et al. (2022) for a modern reference)

**Observation 3.1.** *The expected risk of ridge regression is*

$$R^S(\lambda_1) = \lambda_1^2 \mathbb{E}\left[||M_{\lambda_1}^{(1)} w_1||_{\Sigma_1}^2\right] + \sigma_1^2 \mathbb{E}\left[||M_{\lambda_1}^{(1)} X_1^\top||_{\Sigma_1, F}^2\right].$$

Next we compute the finite sample risk for L2-SP ridge regression.

**Lemma 3.2.** *The expected risk of the transfer estimator decomposes into pure bias, variance induced by the $\beta_0$ prior, and variance induced by estimation error:*

$$R^{TL}(\lambda_1) = B^{TL}(\lambda_1) + \sigma_0^2 V_0^{TL}(\lambda_1) + \sigma_1^2 V_1^{TL}(\lambda_1)$$

*with:*

$$B^{TL}(\lambda_1) = \lambda_1^2 \mathbb{E}\left[||M_{\lambda_1}^{(1)} M_{\lambda_0}^{(0)} X_0^\top X_0 w_0 - M_{\lambda_1}^{(1)} w_1||_{\Sigma_1}^2\right]$$

$$V_0^{TL}(\lambda_1) = \lambda_1^2 \mathbb{E}\left[||M_{\lambda_1}^{(1)} M_{\lambda_0}^{(0)} X_0^\top||_{\Sigma_1, F}^2\right]$$

$$V_1^{TL}(\lambda_1) = \mathbb{E}\left[||M_{\lambda_1}^{(1)} X_1^\top||_{\Sigma_1, F}^2\right].$$

*Sketch.* We expand $\hat{\beta}_1^{\mathrm{TL}} - w_1$ by substituting $y_0 = X_0 w_0 + \epsilon_0$ and $y_1 = X_1 w_1 + \epsilon_1$ into the closed-form expression, isolating a deterministic bias term from two independent noise-driven terms. Since $\epsilon_0$ and $\epsilon_1$ have mean zero and are independent, the cross terms vanish in expectation of the squared $\Sigma_1$-norm, yielding the three-part decomposition.

The following theorem characterizes when transfer helps by comparing the bias introduced by regularizing toward zero (standard ridge, $\hat{\beta}_1^{\mathrm{S}}$) versus regularizing toward the source parameters (L2-SP, $\hat{\beta}_1^{\mathrm{TL}}$). Transfer succeeds when the alignment between the tasks, after filtering through the ridge resolvent and covariance geometry, exceeds a threshold determined by the source estimator's squared norm (how far from zero the anchor is) and the source noise variance $\sigma_0^2$ (how much additional variance the source prior injects). Note that since the Task 1 variance portion of risk is the same for both estimators, it cancels from the comparison.

**Theorem 3.3.** *In the finite sample case with $\lambda_1 > 0$, we gain benefit from transfer learning ($R^{TL}(\lambda_1) < R^S(\lambda_1)$) if and only if:*

$$2\mathbb{E}\left[\langle M_{\lambda_1}^{(1)} M_{\lambda_0}^{(0)} X_0^\top X_0 w_0, M_{\lambda_1}^{(1)} w_1 \rangle_{\Sigma_1}\right]$$
$$> \mathbb{E}\left[||M_{\lambda_1}^{(1)} M_{\lambda_0}^{(0)} X_0^\top X_0 w_0||_{\Sigma_1}^2\right] + \sigma_0^2 \mathbb{E}\left[||M_{\lambda_1}^{(1)} M_{\lambda_0}^{(0)} X_0^\top||_{\Sigma_1, F}^2\right].$$

*Sketch.* The Task 1 variance terms in $R^S$ and $R^{TL}$ are identical and cancel when computing $R^S - R^{TL}$. The remaining difference involves only the bias terms, and expanding the squared $\Sigma_1$-norms yields a cross-term proportional to $\langle \hat{\beta}_0, w_1 \rangle_{\Sigma_1}$; rearranging gives the stated inequality.

As a corollary, in the ridgeless limit we consider when $X_i$ is taken to be isotropic and Gaussian so that the Wishart formulas apply exactly:

**Corollary 3.4.** *In the finite case, if $\Sigma_i = I$ and $\lambda_1 = \lambda_0 = 0$ and $X_i$ is taken to be Gaussian, then $R^{TL}(0) < R^S(0)$ if and only if:*

$$2\langle w_0, w_1 \rangle > ||w_0||^2 + \sigma_0^2 \frac{p}{p - n_0 - 1}.$$

*Sketch.* In the ridgeless isotropic limit, the resolvent $M_\lambda^{(1)}$ converges to the null-space projector $I - P_1$. By isotropy, the projection introduces a common factor of $(p - n_1)/p$ in every term, which cancels from the inequality. The remaining Wishart expectations ($\mathbb{E}[P_0] = n_0/p$ and $\mathbb{E}[\|X_0^+\|_F^2] = n_0/(p - n_0 - 1)$) reduce the condition to the stated scalar inequality, which notably contains no target-side quantities.

A notable feature of the isotropic ridgeless boundary is its complete independence from $n_1$ and $\sigma_1$: the transfer decision depends only on task alignment $\langle w_0, w_1 \rangle$ and source characteristics ($\|w_0\|, \sigma_0, n_0, p$). If transfer outperforms training from scratch with 10 target samples, it also outperforms training from scratch with 10,000 target samples (and the same statement holds across target noise levels). Collecting more target data does not change whether transfer helps in this isotropic regime; only task alignment and source-side quality matter.

We also observe here that the transfer region monotonically shrinks as $\sigma_0$ grows. Thus, in order to maximize the potential transfer benefit it is essential to ensure source task noise is as small as possible.

## 3.2 Deterministic Equivalents and Asymptotics

We will now examine asymptotics for the ridge phase transition identified above and use deterministic equivalents to understand the limiting phase transition. To establish the core deterministic equivalents, we will scale our $\lambda_i$ with $n_i$ and let $\tau_i = \lambda_i/n_i$. We define the following ridge resolvent: Let $S_i := n_i^{-1} X_i^\top X_i$ and $\gamma_i := \lim_{p \to \infty} p/n_i$. Under standard Bai–Silverstein assumptions (e.g. bounded spectral norm of $\Sigma_i$ and weak convergence of its empirical spectral distribution), the resolvent $(S_i + \tau_i I)^{-1}$ admits a deterministic equivalent of the form

$$Q_i(\tau_i) = \left( \tau_i I + \tilde{\delta}_i(\tau_i) \Sigma_i \right)^{-1},$$

where the scalar pair $(\delta_i(\tau_i), \tilde{\delta}_i(\tau_i))$ is the unique positive solution to

$$\delta_i(\tau_i) = \frac{1}{n_i} \operatorname{Tr} \left( \Sigma_i Q_i(\tau_i) \right), \qquad \tilde{\delta}_i(\tau_i) = \frac{1}{1 + \delta_i(\tau_i)}.$$

We will use the notation $A_n \asymp B_n$ to denote deterministic equivalent convergence, meaning that for any sequence of deterministic matrices $D_n$ with uniformly bounded spectral norm, we have $\operatorname{Tr}(D_n(A_n - B_n)) \to 0$ almost surely as $n \to \infty$. This is the standard notion of weak deterministic equivalence used in random matrix theory Bai & Silverstein (2010).

First, the following deterministic equivalents will be useful:

**Observation 3.5.** *Under standard Bai-Silverstein conditions (Bai & Silverstein (2010)), as $p, n_i \to \infty$ with $p/n_i \to \gamma_i$:*

$$n_1 M_{\lambda_1}^{(1)} = (S_1 + \tau_1 I)^{-1} \asymp Q_1(\tau_1),$$

$$\lambda_1 M_{\lambda_1}^{(1)} \asymp \tau_1 Q_1(\tau_1),$$

*and*

$$n_0 M_{\lambda_0}^{(0)} X_0^\top X_0 M_{\lambda_0}^{(0)} \asymp Q_0(\tau_0) - \tau_0 Q_0(\tau_0)^2.$$

Finally, define $t(\tau_0, \tau_1)$ by:

$$\lim_{p \to \infty} p^{-1} \operatorname{Tr} \left( Q_1(\tau_1) \Sigma_1 Q_1(\tau_1) \left( Q_0(\tau_0) - \tau_0 Q_0(\tau_0)^2 \right) \right),$$

Under the Bai-Silverstein assumptions, the eigenvalue distributions of $\Sigma_0$ and $\Sigma_1$ converge weakly and the resolvents $Q_i(\tau_i)$ are uniformly bounded, ensuring that this limit exists and equals the limit of the corresponding trace per dimension. By independence of $X_0$ and $X_1$, we may substitute these deterministic equivalents into the results from Theorem 3.3 to arrive at the following asymptotic decision criterion:

**Theorem 3.6.** *In the asymptotic limit, we gain benefit from transfer learning ($R^{TL}(\lambda_1) < R^S(\lambda_1)$) if and only if:*

$$2 \langle Q_1(\tau_1)(I - \tau_0 Q_0(\tau_0)) w_0, Q_1(\tau_1) w_1 \rangle_{\Sigma_1}$$
$$> \|Q_1(\tau_1)(I - \tau_0 Q_0(\tau_0)) w_0\|_{\Sigma_1}^2 + \sigma_0^2 \gamma_0 t(\tau_0, \tau_1).$$

*Sketch.* By independence of $X_0$ and $X_1$, we replace the random resolvents in Theorem 3.3 with their deterministic equivalents $Q_i(\tau_i)$ from Observation 3.5. The cross terms factor, and all random-matrix quantities reduce to the fixed-point scalar $\tilde{\delta}_i$.

And in the isotropic case:

**Corollary 3.7.** *In the isotropic case where $\Sigma_0 = \Sigma_1 = I$, we (asymptotically) gain benefit from transfer learning ($R^{TL}(\lambda_1) < R^S(\lambda_1)$) if and only if:*

$$2 \langle w_0, w_1 \rangle > (1 - \tau_0 a_0)||w_0||^2 + \sigma_0^2 \gamma_0 a_0,$$

*where $a_0$ is the unique positive solution to*

$$\tau_0 \gamma_0 a_0^2 + (\tau_0 + 1 - \gamma_0)a_0 - 1 = 0.$$

*Sketch.* When $\Sigma_i = I$, the deterministic equivalent becomes scalar: $Q_i(\tau_i) = a_i I$. Substituting into Theorem 3.6, the common factor $a_1^2$ cancels from both sides, eliminating all target-side quantities and leaving a condition involving only $a_0$, $\tau_0$, $\gamma_0$, $\sigma_0$, and $\langle w_0, w_1 \rangle$.

As in the finite sample isotropic ridgeless case, in the isotropic DE (ridge) setting this decision boundary is independent of $\tau_1$, $\gamma_1$, and $\sigma_1$. In particular, whether transfer helps is a source-side question: it does not depend on target sample size, target noise, or target regularization.

As in the finite sample isotropic ridgeless case, alignment of the two tasks must pass a threshold that is independent of other target task characteristics. This transfer benefit region is again monotonically shrinking in $\sigma_0$, and may be maximized by minimizing the quantity on the right. By fixing the overparameterization ratio $\gamma_0 > 1$ we arrive at the following result:

**Theorem 3.8.** *In the asymptotic setting with isotropic data and fixed source model overparameterization $\gamma_0$, for any fixed normalized alignment $\rho = \langle w_0, w_1 \rangle / ||w_0||^2$, there exists a unique source ridge penalty $\tau_0^*$ that maximizes the transfer benefit $\Delta R = R^S - R^{TL}$. Additionally, whenever $\langle w_0, w_1 \rangle \neq ||w_0||^2$, the optimal $\tau_0^*$ for transfer learning differs from the optimal ridge penalty for Task 0 performance.*

*Sketch.* We reparametrize by the shrinkage factor $x = 1 - \tau_0 a_0 \in (0, \gamma_0^{-1})$, which is monotonically related to $\tau_0$. The transfer risk becomes a function $J(x)$ whose first-order condition is $G(x^*) = \langle w_0, w_1 \rangle$, where

$$G(x) = x||w_0||^2 + \frac{\sigma_0^2 \gamma_0}{2} \frac{x(2 - \gamma_0 x)}{(1 - \gamma_0 x)^2}$$

is strictly increasing on $(0, \gamma_0^{-1})$. Since the source-optimal shrinkage satisfies $G(x_S^*) = ||w_0||^2$, the two optima coincide if and only if $\langle w_0, w_1 \rangle = ||w_0||^2$, i.e., $\rho = 1$. Thus the transfer-optimal $\tau_0^*$ is implicitly determined by $G(x^*) = \rho ||w_0||^2$ followed by inversion of the $x \mapsto \tau_0$ bijection.

As a corollary, we identify that there is a consistent relationship between transfer-optimal and source-optimal regularization.

**Corollary 3.9.** *Let $\tau_0^*$ be the transfer-optimal regularization penalty and $\tau_S^*$ be the source-optimal regularization penalty.*

- *If the tasks are imperfectly aligned ($0 < \rho < 1$), then $\tau_0^* > \tau_S^*$ (transfer requires stronger regularization).*

- *If the tasks are super-aligned ($\rho > 1$), then $\tau_0^* < \tau_S^*$ (transfer requires weaker regularization).*

*This phase transition depends solely on task alignment and holds for all noise levels $\sigma_0 > 0$.*

*Sketch.* Since $G$ is strictly increasing, $\rho < 1$ implies $G(x^*) < G(x_S^*)$, so $x^* < x_S^*$. Because $x$ is strictly decreasing in $\tau_0$, smaller shrinkage corresponds to larger regularization, giving $\tau_0^* > \tau_S^*$. The argument reverses for $\rho > 1$.

In particular, this shows that when the tasks are imperfectly aligned ($0 < \rho < 1$), the transfer-optimal solution always uses more regularization than one would typically use for the source task alone.

*Remark* 3.10 (Non-isotropic extension). Theorems 3.8 and Corollary 3.9 are stated for the isotropic case, where the scalar deterministic equivalents make the optimization tractable. In the non-isotropic setting (Theorem 3.6), the transfer risk depends on $\tau_0$ through the matrix-valued resolvent $Q_0(\tau_0)$ and the optimization is no longer one-dimensional. The same mathematical tools apply in principle (the monotonicity of the resolvent in $\tau_0$ is preserved), but the optimal $\tau_0^*$ depends on the alignment of $w_0$ and $w_1$ relative to the eigenbasis of $\Sigma_0$, rather than on a single scalar $\rho$. We probe the non-isotropic case experimentally in Section 4.

## 4 Empirical Validation

We validate the phase transition predictions with a controlled synthetic experiment and then study nonlinear networks on MNIST, CIFAR-10, and 20 Newsgroups to test whether the mismatch persists beyond linear models.

### 4.1 Synthetic Validation of Phase Transition

#### 4.1.1 Experimental Setup

To verify the alignment-dependent phase transition predicted by Corollary 3.9, we conduct a controlled synthetic experiment using the generative model defined in Section 2. We generate isotropic data ($\Sigma_0 = \Sigma_1 = I$) with $p = 500$, $n_{source} = 250$ ($\gamma_0 = 2.0$), and $n_{target} = 50$. We fix the source noise $\sigma_0^2 = 1.0$ and target noise $\sigma_1^2 = 0.1$.

We sweep the task alignment $\rho = \langle w_0, w_1 \rangle / ||w_0||^2$ from 0.5 to 1.5, covering both regimes. For each alignment level, we train source ridge models over a logarithmically spaced grid of regularization strengths, identify the source-optimal $\lambda_S^*$ minimizing source risk, compute the L2-SP transfer estimator on the target task (with fixed $\lambda_1 = 0.1$), identify the transfer-optimal $\lambda_{TL}^*$, and report the ratio $\lambda_{TL}^*/\lambda_S^*$. Results are averaged over 10 seeds.

#### 4.1.2 Results

Figure 1a displays the ratio $\lambda_{TL}^*/\lambda_S^*$ as a function of task alignment, sweeping $\gamma_0 \in \{1.5, 2.0, 3.0, 5.0\}$. The results match the theoretical predictions across all $\gamma_0$ values. For imperfect alignment ($\rho < 1$), the ratio is consistently $> 1$, confirming that transfer requires stronger regularization (e.g., $\sim 1.5\times$ at $\gamma_0 = 2.0$, $\rho = 0.8$). For super-alignment ($\rho > 1$), the ratio drops below 1 ($\sim 0.7\times$ at $\rho = 1.2$). The crossover occurs precisely at $\rho = 1.0$ for all $\gamma_0$, and the magnitude of the effect increases with $\gamma_0$, as higher overparameterization amplifies the gap between $\tau_0^*$ and $\tau_S^*$.

We additionally probe the non-isotropic case (Theorem 3.6) by replacing the identity covariance with a power-law spectrum ($\Sigma$ diagonal with eigenvalues $k^{-\alpha}$, normalized so that $\text{Tr}(\Sigma)/p = 1$). Figure 1b shows results for $\alpha \in \{0.5, 1.0, 2.0\}$ alongside the isotropic baseline. For mild spectral decay ($\alpha = 0.5$), the qualitative pattern is preserved. As the spectrum concentrates ($\alpha = 2.0$), a small number of eigenvalues dominate and the scalar $\rho$ no longer cleanly predicts the transition boundary. This is consistent with Theorem 3.6, which predicts that the boundary depends on the alignment of $w_0, w_1$ relative to the eigenbasis of $\Sigma$.

### 4.2 Nonlinear Transfer Learning Experiments

#### 4.2.1 Experimental Setup

**MNIST.** We use a 2-layer MLP ($784 \rightarrow 128 \rightarrow 64 \rightarrow 5$) and split digits into source (round digits: 0,3,6,8,9) and target (angular digits: 1,2,4,5,7). The target training set is subsampled to 10% to create a realistic limited-data transfer scenario.

**CIFAR-10.** We use a small CNN (two conv layers, one FC layer, $\sim$530K parameters) and split classes into source (animals: bird, cat, deer, dog, frog) and target (vehicles+: airplane, automobile, horse, ship, truck). The target training set is subsampled to 5%.

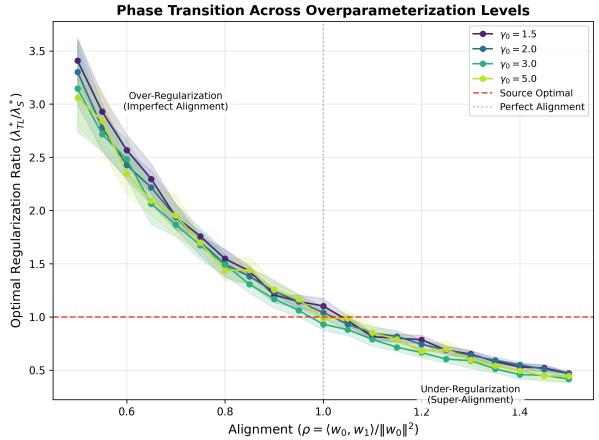 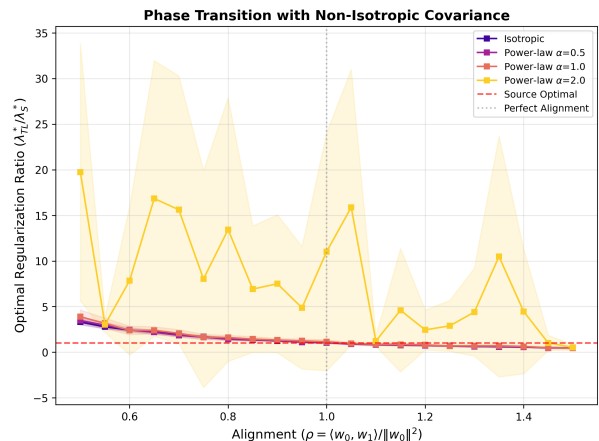

(a) Phase transition across overparameterization levels $\gamma_0 \in \{1.5, 2.0, 3.0, 5.0\}$. The transition at $\rho = 1$ is robust.

(b) Phase transition under power-law covariance spectra ($\alpha \in \{0.5, 1.0, 2.0\}$) versus isotropic baseline.

Figure 1: Robustness of the phase transition. (a) The transition persists across overparameterization levels, with divergence magnitude increasing with $\gamma_0$. (b) Under power-law covariance, the pattern holds for mild decay but becomes noisier as the spectrum concentrates.

**20 Newsgroups.** We use TF-IDF features (5000 dimensions) with a 2-layer MLP (5000→256→128→num_classes) and split categories into source (tech: comp.\*, sci.\*) and target (non-tech: rec.\*, talk.\*, misc.\*, alt.atheism). The target training set is subsampled to 10%.

For all experiments, source models are trained with Adam (learning rate $10^{-3}$, batch size 64) over 19 logarithmically spaced weight decay values from $10^{-6}$ to $10^{-1}$ plus zero. Transfer initializes all layers from the source model, reinitializes the classification head, and fine-tunes with Adam (learning rate $10^{-3}$, weight decay $10^{-4}$, batch size 64). We also run an explicit L2-SP baseline with penalty $\frac{\alpha}{2}\|\theta - \theta_0\|^2$ ($\alpha = 10^{-3}$). Results are averaged over 10 seeds.

### 4.2.2 Results

Figure 2 shows that across all three domains, transfer-optimal performance occurs at stronger source regularization than source-optimal performance, consistent with the over-regularization regime predicted by our theory. In MNIST, source accuracy peaks at near-zero weight decay (98.5%) while transfer peaks at WD $\approx$ 3–7 $\times 10^{-3}$, a gap of approximately three orders of magnitude. In CIFAR-10, source accuracy peaks at WD $\approx 3 \times 10^{-3}$ (71.2%) while transfer peaks at $\approx 7 \times 10^{-3}$ (74.7%). In 20 Newsgroups, source accuracy peaks near zero WD (67.5%) while transfer peaks around $10^{-3}$.

Across all three experiments, the explicit L2-SP baseline produces nearly identical behavior to standard SGD fine-tuning: the optimal source weight decay and accuracy curves closely overlap under both protocols, supporting the view that SGD fine-tuning acts as an implicit form of L2-SP regularization. Figure 3 additionally shows that fine-tuned parameters remain close to the source initialization throughout training, with more strongly regularized source models exhibiting smaller parameter drift, a necessary condition for the L2-SP analogy.

### 4.2.3 Target-Side Independence Ablation

Corollary 3.9 predicts that whether transfer helps depends only on source-side quantities and alignment, not on target sample size or noise. To test this, we run the MNIST and CIFAR-10 transfer experiments across target training fractions $\{1\%, 5\%, 10\%, 20\%, 50\%\}$. Figure 4 shows that in both cases the transfer-optimal source weight decay is stable across all fractions: WD $\approx 7 \times 10^{-3}$ for MNIST (accuracy ranging from 85.6% to 98.4%) and WD $\approx 7 \times 10^{-3}$ for CIFAR-10 (accuracy ranging from 60.7% to 84.7%). The absolute accuracy

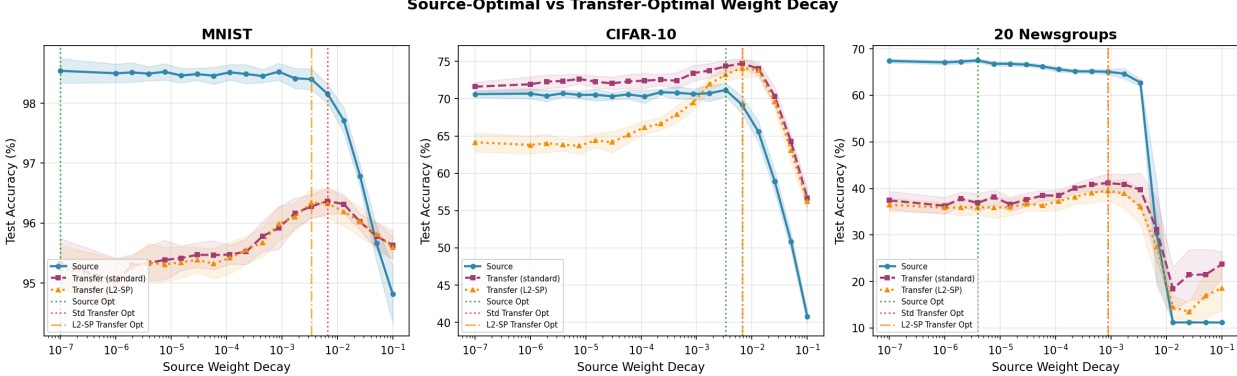

Figure 2: Source-optimal versus transfer-optimal weight decay across three benchmarks (19 WD values, 10 seeds). Solid blue: source accuracy; dashed red: standard transfer; dotted orange: L2-SP transfer. Vertical lines mark optima. Transfer-optimal WD exceeds source-optimal WD in all cases, and L2-SP closely tracks standard transfer.

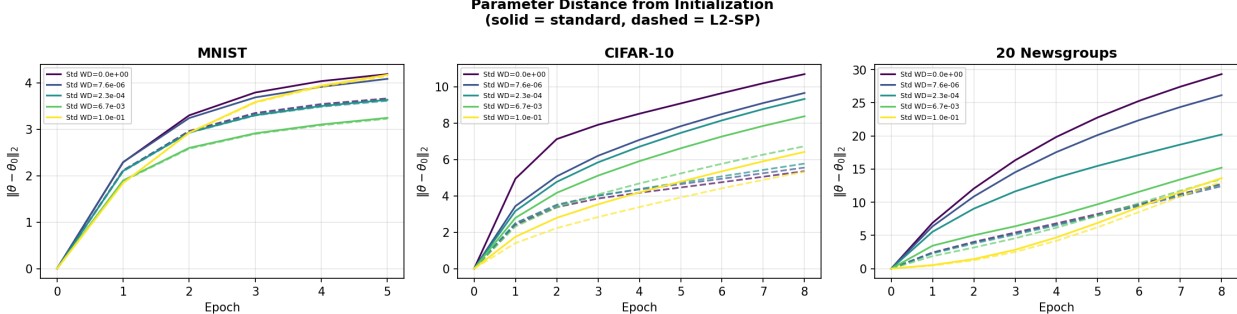

Figure 3: Parameter distance $\|\theta - \theta_0\|_2$ during fine-tuning, grouped by source weight decay. More strongly regularized source models produce parameters that remain closer to initialization.

increases with more target data, but the *choice of source regularization* is essentially independent of target sample size, as the theory predicts.

## 5   Conclusion

We have proven a fundamental misalignment in transfer learning: training a source model to minimize its own risk is generically suboptimal for maximizing transfer benefit. This source-optimal versus transfer-optimal divergence implies that source regularization should be chosen with the downstream objective in mind and depends on task alignment. We provide explicit transfer-versus-scratch boundaries for L2-SP ridge that hold at finite $p, n$ and extend to general covariance via deterministic equivalents. In isotropic limits, whether transfer helps is independent of target sample size and noise: if transfer helps with 10 target samples, it helps with 10,000. Optimizing source regularization for target performance yields a unique $\tau_0^*$ that differs generically from the source-task-optimal choice.

Our experiments on MNIST, CIFAR-10, and 20 Newsgroups show that the source-optimal versus transfer-optimal mismatch persists beyond the linear setting. Explicit L2-SP fine-tuning closely tracks standard SGD, parameter tracking confirms proximity to the source initialization, and the target-side independence prediction is confirmed on both MNIST and CIFAR-10: the transfer-optimal source weight decay is constant across target fractions spanning a $50\times$ range. In all nonlinear regimes we tested, transfer consistently prefers stronger source regularization. We did not observe a super-aligned regime, suggesting that the under-regularization phase may be difficult to realize in deep-learning pipelines where feature learning plays a significant role.

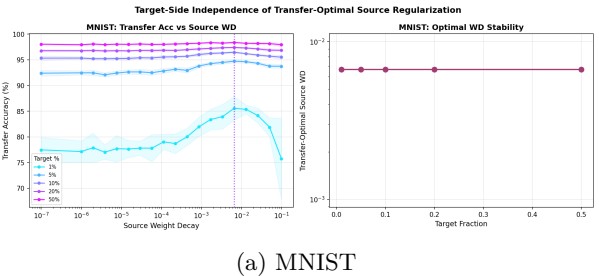 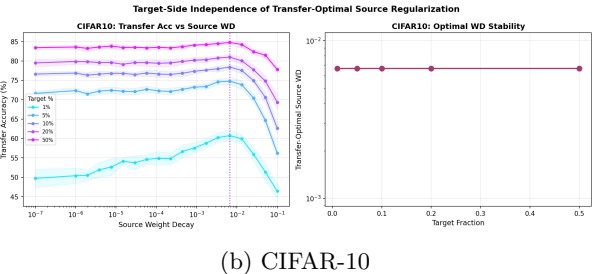

|(a) MNIST|(b) CIFAR-10|

Figure 4: Target-side independence ablation. The transfer-optimal source WD (vertical lines) is stable at $\approx 7 \times 10^{-3}$ across target training fractions from 1% to 50% for both MNIST and CIFAR-10.

## 5.1 Practical Implications

Our results provide several concrete insights for practitioners. First, minimizing source noise is critical for maximizing positive transfer, as the transfer benefit shrinks monotonically with $\sigma_0^2$. Second, the independence of the transfer decision from target sample size (confirmed by the MNIST and CIFAR-10 ablations in Figure 4) simplifies model selection when target data availability is uncertain.

The phase transition provides actionable guidance: under imperfect alignment ($\langle w_0, w_1 \rangle < ||w_0||^2$), one should increase regularization beyond what is optimal for the source task alone. That over-regularization can improve transfer is consistent with prior findings Xuhong et al. (2018), but our contribution is making this intuition analytically precise, characterizing the exact divergence as a function of alignment and noise (e.g., ~50% at $\rho = 0.8$; see Figure 1a). We note that the theory does not assume $\rho$ is known at pretraining time; rather, it explains *why* over-regularization is a robust strategy (it is optimal across the entire imperfect alignment regime) and quantifies *how much* the two optima diverge.

While we are cautious about directly extrapolating the precise functional forms to foundation-model-scale pipelines, the underlying mechanism (that source-optimal training over-fits to the source signal direction, leaving a suboptimal anchor for transfer) is not specific to the linear setting, and our nonlinear experiments consistently confirm the over-regularization pattern.

## 5.2 Scope and Future Directions

The precise functional forms we derive are specific to linear ridge regression. Our synthetic ablations confirm robustness to variations in $\gamma_0$ and mild spectral decay, though the scalar $\rho$ becomes less predictive as the spectrum concentrates (consistent with Theorem 3.6). Our nonlinear experiments suggest the mismatch persists in deep networks across vision and text, while the super-aligned regime appears difficult to realize in standard deep-learning setups. We assume Gaussian noise and employ either exact isotropy or Bai-Silverstein conditions; extensions to more general noise models and covariance structures remain open problems.

We focus on the L2-SP protocol with Euclidean penalties, which differs from methods using task-specific metrics or adaptive importance weights. L2-SP also represents a closed-form solution, while practical deep learning relies on iterative SGD. The implicit regularization of early-stopped SGD typically keeps parameters close to initialization, qualitatively matching L2-SP, though for networks with very flat loss landscapes SGD may travel far from initialization, potentially weakening the analogy. The recent work of Ghane et al. Ghane et al. (2024) provides a complementary perspective, deriving universal asymptotics for gradient-descent fine-tuning from pretrained anchors. Their results characterize the iterative (GD) regime while ours address the closed-form L2-SP solution; together these bracket practical fine-tuning. Understanding when the closed-form risk accurately predicts iterative fine-tuning behavior remains an important open question.

**Super-alignment in nonlinear networks.** Our theory predicts a phase reversal at $\rho = 1$: for $\rho > 1$, transfer benefits from *weaker* source regularization. This is clearly visible in the synthetic experiment (Figure 1a), but in all nonlinear experiments we conducted (including extensive search across task splits and architectures) we did not observe a super-aligned regime. In deep networks, feature learning causes

the effective task representations to co-adapt during training, making it structurally unlikely for the target projection to exceed the source norm. Standard classification splits with disjoint label sets also produce imperfect alignment by construction. The absence of super-alignment in practice strengthens the practical takeaway: over-regularization is the relevant regime.

**Practical alignment estimation.** While $\rho$ as defined in the linear setting is not directly computable for deep networks, proxy measures such as gradient alignment, representation similarity (e.g., CKA), or task2vec embeddings could serve as surrogates. The key qualitative insight (the distinction between imperfect and super-alignment determines the direction of regularization adjustment) holds even when exact estimation of $\rho$ is infeasible.

**Multi-target settings.** Our results focus on two-task transfer, but many practical scenarios involve multiple downstream tasks. Extending to a multi-target setting introduces a minimax or average-case optimization problem where the source regularization must balance alignment with a distribution of targets. Whether a single $\tau_0^*$ can serve all targets or whether Pareto-optimal trade-offs emerge is a natural direction for future work. More broadly, connecting these findings to gradient-descent dynamics and investigating whether early stopping, learning rate schedules, or architectural choices can implicitly achieve transfer-optimal regularization remain important directions for practice.

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

## A    Detailed Proofs

Here we will re-state and prove the relevant Lemmas and Theorems mentioned above.

### A.1    Finite Sample Risk Formulas

We begin by re-writing our L2-SP transfer learning estimator for the reader's convenience:

$$\hat{\beta}_1^{\mathrm{TL}}(\lambda_1 \mid \hat{\beta}_0) = (X_1^\top X_1 + \lambda_1 I)^{-1}\left(X_1^\top y_1 + \lambda_1 \hat{\beta}_0(\lambda_0)\right).$$

We additionally note that expected risk can be computed by:

$$R(\beta) = E\left[\|\beta - w_1\|_{\Sigma_1}^2\right].$$

We can now prove the first important Lemma:

**Lemma 3.2** The expected risk of the transfer estimator decomposes into pure bias, variance induced by the $\beta_0$ prior, and variance induced by estimation error:

$$R^{TL}(\lambda_1) = B^{TL}(\lambda_1) + \sigma_0^2 V_0^{TL}(\lambda_1) + \sigma_1^2 V_1^{TL}(\lambda_1)$$

with:

$$B^{TL}(\lambda_1) = \lambda_1^2 \mathbb{E}\left[\|M_{\lambda_1}^{(1)} M_{\lambda_0}^{(0)} X_0^\top X_0 w_0 - M_{\lambda_1}^{(1)} w_1\|_{\Sigma_1}^2\right]$$

$$V_0^{TL}(\lambda_1) = \lambda_1^2 \mathbb{E}\left[\|M_{\lambda_1}^{(1)} M_{\lambda_0}^{(0)} X_0^\top\|_{\Sigma_1,F}^2\right]$$

$$V_1^{TL}(\lambda_1) = \mathbb{E}\left[\|M_{\lambda_1}^{(1)} X_1^\top\|_{\Sigma_1,F}^2\right].$$

*Proof.* We verify the decomposition as follows. First note:

$$\hat{\beta}_1^{\mathrm{TL}}(\lambda \mid \hat{\beta}_0) - w_1 = \lambda M_\lambda^{(1)} \hat{\beta}_0^S(\lambda_0) + \hat{\beta}_1^S(\lambda) - w_1$$

$$= \lambda M_\lambda^{(1)} M_{\lambda_0}^{(0)} X_0^\top y_0 + M_\lambda^{(1)} X_1^\top y_1 - w_1$$

$$= \lambda M_\lambda^{(1)} M_{\lambda_0}^{(0)} X_0^\top X_0 w_0 + \lambda M_\lambda^{(1)} M_{\lambda_0}^{(0)} X_0^\top \epsilon_0 + M_\lambda^{(1)} X_1^\top X_1 w_1 + M_\lambda^{(1)} X_1^\top \epsilon_1 - w_1$$

$$= \left(\lambda M_\lambda^{(1)} M_{\lambda_0}^{(0)} X_0^\top X_0 + \left(M_\lambda^{(1)} X_1^\top X_1 - I\right)\right) w_1 + \left(\lambda M_\lambda^{(1)} M_{\lambda_0}^{(0)} X_0^\top\right) \epsilon_0 + \left(M_\lambda^{(1)} X_1^\top\right) \epsilon_1.$$

We then take expectation of the squared $\Sigma_1$ norm of the above expression and since $\epsilon_0, \epsilon_1$ are taken independently with mean 0, the cross terms cancel and we are left with the desired result. $\square$

We now remind the reader that the expected risk of the standard Ridge estimator is

$$R^S(\lambda_1) = \lambda_1^2 \mathbb{E}\left[\|M_{\lambda_1}^{(1)} w_1\|_{\Sigma_1}^2\right] + \sigma_1^2 \mathbb{E}\left[\|M_{\lambda_1}^{(1)} X_1^\top\|_{\Sigma_1,F}^2\right].$$

We can therefore compare these quantities to identify when we expect Transfer to outperform training from scratch.

**Theorem 3.3**. In the finite sample case with $\lambda_1 > 0$, we gain benefit from transfer learning ($R^{TL}(\lambda_1) < R^S(\lambda_1)$) if and only if:

$$2\mathbb{E}\left[\langle M_{\lambda_1}^{(1)} M_{\lambda_0}^{(0)} X_0^\top X_0 w_0, M_{\lambda_1}^{(1)} w_1 \rangle_{\Sigma_1}\right] > \mathbb{E}\left[\|M_{\lambda_1}^{(1)} M_{\lambda_0}^{(0)} X_0^\top X_0 w_0\|_{\Sigma_1}^2\right] + \sigma_0^2 \mathbb{E}\left[\|M_{\lambda_1}^{(1)} M_{\lambda_0}^{(0)} X_0^\top\|_{\Sigma_1,F}^2\right].$$

*Proof.* First we notice the $\sigma_1$ terms of $R^S(\lambda_1)$ and $R^{TL}(\lambda_1|\beta_0)$ exactly cancel. We are therefore left with

$$R^S(\lambda) - R^{TL}(\lambda) = \mathbb{E}\left[||(M_\lambda^{(1)} X_1^\top X_1 - I)w_1||_{\Sigma_1}^2\right] - \mathbb{E}\left[||\lambda M_\lambda^{(1)} M_{\lambda_0}^{(0)} X_0^\top X_0 w_0 + \left(M_\lambda^{(1)} X_1^\top X_1 - I\right)w_1||_{\Sigma_1}^2\right]$$

$$-\sigma_0^2 \lambda^2 \mathbb{E}\left[||M_\lambda^{(1)} M_{\lambda_0}^{(0)} X_0^\top ||_{\Sigma_1,F}^2\right]$$

$$= -\mathbb{E}\left[||\lambda M_\lambda^{(1)} M_{\lambda_0}^{(0)} X_0^\top X_0 w_0||_{\Sigma_1}^2\right] - 2\lambda \mathbb{E}\left[\langle (M_\lambda^{(1)} X_1^\top X_1 - I)w_1, M_\lambda^{(1)} M_{\lambda_0}^{(0)} X_0^\top X_0 w_0 \rangle_{\Sigma_1}\right]$$

$$-\sigma_0^2 \lambda^2 \mathbb{E}\left[||M_\lambda^{(1)} M_{\lambda_0}^{(0)} X_0^\top ||_{\Sigma_1,F}^2\right].$$

We note here that $(M_\lambda^{(1)} X_1^\top X_1 - I) = -\lambda M_\lambda^{(1)}$ and thus we have

$$\mathbb{E}\left[\langle (M_\lambda^{(1)} X_1^\top X_1 - I)w_1, M_\lambda^{(1)} M_{\lambda_0}^{(0)} X_0^\top X_0 w_0 \rangle_{\Sigma_1}\right] = -\lambda \mathbb{E}\left[\langle M_\lambda^{(1)} w_1, M_\lambda^{(1)} M_{\lambda_0}^{(0)} X_0^\top X_0 w_0 \rangle_{\Sigma_1}\right].$$

After some algebra and canceling the common $\lambda^2$ terms, we arrive at our desired inequality. $\square$

**Corollary 3.4** In the finite case, if $\Sigma_i = I$ and $\lambda_1 = \lambda_0 = 0$, and $X_i$ is taken to be Gaussian, then $R^{TL}(0) < R^S(0)$ if and only if:

$$2\langle w_0, w_1 \rangle > ||w_0||^2 + \sigma_0^2 \frac{p}{p - n_0 - 1}.$$

*Proof.* We first note

$$\lim_{\lambda_1 \searrow 0} M_{\lambda_1}^{(1)} = \lim_{\lambda \searrow 0}(X_1^\top X_1 + \lambda I)^{-1} = (X_1^\top X_1)^+ = I - X_1^+ X_1.$$

By conditioning on $X_0$ and taking expectation with respect to $X_1$, the projection $I - X_1^+ X_1$ introduces a common factor of $\frac{p-n_1}{p}$ to all terms involving $w_1$ or the $X_0$-dependent prior (due to the isotropy of $X_1$). Specifically, for any fixed vector $v$, $\mathbb{E}_{X_1}[||(I - X_1^+ X_1)v||^2] = \frac{p-n_1}{p}||v||^2$. We can thus eliminate the target projection effects from the expression. We are then left with:

$$2\mathbb{E}\left[\langle M_{\lambda_0}^{(0)} X_0^\top X_0 w_0, w_1 \rangle\right] > \mathbb{E}\left[||M_{\lambda_0}^{(0)} X_0^\top X_0 w_0||_{\Sigma_1}^2\right] + \sigma_0^2 \mathbb{E}\left[||M_{\lambda_0}^{(0)} X_0^\top ||_{\Sigma_1,F}^2\right].$$

We now note that

$$\lim_{\lambda_0 \searrow 0} M_{\lambda_0}^{(0)} X_0^\top = X_0^+,$$

and thus we have

$$2\mathbb{E}\left[\langle X_0^+ X_0 w_0, w_1 \rangle\right] > \mathbb{E}\left[||X_0^+ X_0 w_0||_{\Sigma_1}^2\right] + \sigma_0^2 \mathbb{E}\left[||X_0^+||_{\Sigma_1,F}^2\right].$$

We note here that since $\Sigma_0 = I$ we have $E\left[X_0^+ X_0\right] = \frac{n_0}{p} I$. Additionally, $E\left[||X_0^+||_F^2\right] = \frac{n_0}{p - n_0 - 1}$. We can now take expectation to see

$$2\frac{n_0}{p}\langle w_0, w_1 \rangle > \frac{n_0}{p}||w_0||^2 + \sigma_0^2 \frac{n_0}{p - n_0 - 1}.$$

By multiplying by $\frac{p}{n_0}$ we arrive at the desired result. $\square$

## A.2 Deterministic Equivalents and Asymptotics

We will now examine asymptotics for the ridge phase transition identified above and use deterministic equivalents to understand the limiting phase transition. To establish the core deterministic equivalents, we must first let $\tau_i = \lambda_i / n_i$, and we define the following ridge resolvent: Let $S_i := n_i^{-1} X_i^\top X_i$ and $\gamma_i := \lim_{p \to \infty} p/n_i$. Under standard Bai–Silverstein assumptions, the resolvent $(S_i + \tau_i I)^{-1}$ admits a deterministic equivalent

$$Q_i(\tau_i) = \left(\tau_i I + \tilde{\delta}_i(\tau_i)\Sigma_i\right)^{-1},$$

where the scalar pair $(\delta_i(\tau_i), \tilde{\delta}_i(\tau_i))$ is the unique positive solution to

$$\delta_i(\tau_i) = \frac{1}{n_i} \operatorname{Tr}\left(\Sigma_i Q_i(\tau_i)\right), \qquad \tilde{\delta}_i(\tau_i) = \frac{1}{1 + \delta_i(\tau_i)}.$$

**Observation 3.5** Under standard Bai-Silverstein conditions (Bai & Silverstein (2010)), as $p, n_i \to \infty$ with $p/n_i \to \gamma_i$:

$$n_1 M_{\lambda_1}^{(1)} = (S_1 + \tau_1 I)^{-1} \asymp Q_1(\tau_1),$$

$$\lambda_1 M_{\lambda_1}^{(1)} \asymp \tau_1 Q_1(\tau_1),$$

and

$$n_0 M_{\lambda_0}^{(0)} X_0^\top X_0 M_{\lambda_0}^{(0)} \asymp Q_0(\tau_0) - \tau_0 Q_0(\tau_0)^2.$$

*Proof.* These equivalences follow from standard results in Bai & Silverstein (2010). Specifically, under the Bai-Silverstein conditions, the empirical resolvent $(X_i^\top X_i / n_i + \tau_i I)^{-1}$ admits the deterministic equivalent $Q_i(\tau_i) = (\tau_i I + \tilde{\delta}_i(\tau_i)\Sigma_i)^{-1}$, where $(\delta_i(\tau_i), \tilde{\delta}_i(\tau_i))$ is the unique positive solution pair to the Silverstein fixed-point equations. By dividing by $\lambda_1$ we arrive at the first deterministic equivalence.

On the sample side we know

$$M_{\lambda_0}^{(0)} X_0^\top = X_0^\top M_{\lambda_0}^{(0)} = n_0^{-1} X_0^\top \left(X_0^\top X_0 / n_0 + \lambda_0 I\right)^{-1},$$

and the second deterministic equivalent follows.

Finally we use the following fact:

$$M_{\lambda_0}^{(0)} X_0^\top X_0 M_{\lambda_0}^{(0)} = M_{\lambda_0}^{(0)} \left((X_0^\top X_0 + \lambda_0 I) - \lambda_0 I\right) M_{\lambda_0}^{(0)} = M_{\lambda_0}^{(0)} - \lambda_0 (M_{\lambda_0}^{(0)})^2.$$

From this it is easy to see the third deterministic equation holds. $\qquad \square$

Finally, define $t(\tau_0, \tau_1)$ by:

$$\lim_{p \to \infty} p^{-1} \operatorname{Tr}\left(Q_1(\tau_1)\Sigma_1 Q_1(\tau_1)\left(Q_0(\tau_0) - \tau_0 Q_0(\tau_0)^2\right)\right),$$

which we know exists under the Bai-Silverstein assumptions. By independence of $X_0$ and $X_1$, we may substitute these in to the results from Theorem 3.3 to arrive at the following asymptotic decision criterion:

**Theorem 3.6** In the asymptotic limit, we gain benefit from transfer learning $(R^{TL}(\lambda_1) < R^S(\lambda_1))$ if and only if:

$$2\left\langle Q_1(\tau_1)(I - \tau_0 Q_0(\tau_0))w_0, Q_1(\tau_1)w_1 \right\rangle_{\Sigma_1} > ||Q_1(\tau_1)(I - \tau_0 Q_0(\tau_0))w_0||_{\Sigma_1}^2 + \sigma_0^2 \gamma_0 t(\tau_0, \tau_1).$$

*Proof.* This result follows directly from substituting the deterministic equivalents from Observation 3.5 into the results from Theorem 3.3 combined with the independence of $X_1$ and $X_0$. $\qquad \square$

**Corollary 3.7** In the isotropic case where $\Sigma_0 = \Sigma_1 = I$, we (asymptotically) gain benefit from transfer learning $(R^{TL}(\lambda_1) < R^S(\lambda_1))$ if and only if:

$$2\left\langle w_0, w_1 \right\rangle > (1 - \tau_0 a_0)||w_0||^2 + \sigma_0^2 \gamma_0 a_0,$$

where $a_0$ is the unique positive solution to

$$\tau_0 \gamma_0 a_0^2 + (\tau_0 + 1 - \gamma_0)a_0 - 1 = 0.$$

*Proof.* We note that when $\Sigma_i = I$, the deterministic equivalent is scalar: $Q_i(\tau_i) = a_i I$ where $a_i$ is the unique positive solution to

$$\tau_i \gamma_i a_i^2 + (\tau_i + 1 - \gamma_i) a_i - 1 = 0.$$

We can substitute this in to the relevant quantities from Theorem 3.6 to see:

$$2 \langle Q_1(\tau_1)(I - \tau_0 Q_0(\tau_0)) w_0, Q_1(\tau_1) w_1 \rangle_{\Sigma_1} = 2 a_1^2 (1 - \tau_0 a_0) \langle w_0, w_1 \rangle \text{ and}$$

$$||Q_1(\tau_1)(I - \tau_0 Q_0(\tau_0)) w_0||_{\Sigma_1}^2 = a_1^2 (1 - \tau_0 a_0)^2 ||w_0||^2.$$

We now examine

$$\begin{aligned}
t(\tau_0, \tau_1) &= \lim_{p \to \infty} p^{-1} Tr \left( a_1^2 I (a_0 I - \tau_0 a_0^2 I) \right) \\
&= a_1^2 Tr \left( a_0 I - \tau_0 a_0^2 I \right) / p \\
&= a_1^2 (a_0 - \tau_0 a_0^2) \\
&= a_1^2 a_0 (1 - \tau_0 a_0).
\end{aligned}$$

Substituting into Theorem 3.6 and canceling the common factor $a_1^2$, we obtain

$$2(1 - \tau_0 a_0) \langle w_0, w_1 \rangle > (1 - \tau_0 a_0)^2 ||w_0||^2 + \sigma_0^2 \gamma_0 a_0 (1 - \tau_0 a_0).$$

Dividing by $(1 - \tau_0 a_0) > 0$ yields the stated condition. $\qquad \square$

**Theorem 3.8** In the asymptotic setting with isotropic data and fixed source model overparameterization $\gamma_0$, for any fixed normalized alignment $\rho = \langle w_0, w_1 \rangle / ||w_0||^2$, there exists a unique source ridge penalty $\tau_0^*$ that maximizes the transfer benefit $\Delta R = R^S - R^{TL}$. Additionally, outside of a measure-zero set of parameters, the optimal $\tau_0^*$ for transfer learning differs from the optimal ridge penalty for Task 0 performance.

*Proof.* We seek to maximize the asymptotic risk benefit $\Delta R(\tau_0) = R^S - R^{TL}(\tau_0)$. Since $R^S$ is independent of $\tau_0$, this is equivalent to minimizing $R^{TL}(\tau_0)$. Using the isotropic deterministic equivalents derived in Corollary 3.7, the transfer risk is proportional to:

$$J(\tau_0) = (1 - \tau_0 a_0)^2 ||w_0||^2 + \sigma_0^2 \gamma_0 a_0 (1 - \tau_0 a_0) - 2(1 - \tau_0 a_0) \langle w_0, w_1 \rangle,$$

where $a_0$ is implicitly defined by $\tau_0$. Let $x(\tau_0) = 1 - \tau_0 a_0$. From the fixed-point equation $\tau_0 \gamma_0 a_0^2 + (\tau_0 + 1 - \gamma_0) a_0 - 1 = 0$, we can derive the bijection $x = \frac{a_0}{1 + \gamma_0 a_0}$. Note that $a_0$ is strictly decreasing in $\tau_0$ (from $\infty$ to 0), and $x$ is strictly increasing in $a_0$ (mapping $(0, \infty)$ to $(0, \gamma_0^{-1})$). Thus, optimizing with respect to $\tau_0$ is equivalent to optimizing with respect to the shrinkage factor $x$. Substituting $a_0 = \frac{x}{1 - \gamma_0 x}$ into the objective:

$$J(x) = x^2 ||w_0||^2 + \frac{\sigma_0^2 \gamma_0 x^2}{1 - \gamma_0 x} - 2x \langle w_0, w_1 \rangle.$$

The derivative with respect to $x$ is:

$$J'(x) = 2x ||w_0||^2 + \sigma_0^2 \gamma_0 \frac{2x(1 - \gamma_0 x) + \gamma_0 x^2}{(1 - \gamma_0 x)^2} - 2 \langle w_0, w_1 \rangle = 2x ||w_0||^2 + \sigma_0^2 \gamma_0 \frac{2x - \gamma_0 x^2}{(1 - \gamma_0 x)^2} - 2 \langle w_0, w_1 \rangle.$$

Let $G(x) = x ||w_0||^2 + \frac{\sigma_0^2 \gamma_0}{2} \frac{x(2 - \gamma_0 x)}{(1 - \gamma_0 x)^2}$. For $x \in (0, \gamma_0^{-1})$, $G(x)$ is strictly increasing. The optimal shrinkage $x^*$ is the unique solution to $G(x^*) = \langle w_0, w_1 \rangle$. Define the source-optimal shrinkage $x_S^*$ as the minimizer of the source risk (which corresponds to setting $w_1 = w_0$ in the transfer risk). Thus $x_S^*$ satisfies $G(x_S^*) = ||w_0||^2$. Comparing the two conditions:

$$G(x^*) = \langle w_0, w_1 \rangle \quad \text{and} \quad G(x_S^*) = ||w_0||^2.$$

Since $G$ is strictly increasing/injective, $x^* = x_S^*$ if and only if $\langle w_0, w_1 \rangle = ||w_0||^2$, i.e., $\rho = 1$. Thus the set of parameters where $\tau_0^* = \tau_S^*$ is precisely $\{\rho = 1\}$, which has Lebesgue measure zero in the space of task vectors $(w_0, w_1)$. Outside this set, $\tau_0^* \neq \tau_S^*$. $\qquad \square$

**Corollary 3.9** Let $\tau_0^*$ be the transfer-optimal regularization penalty and $\tau_S^*$ be the source-optimal regularization penalty.

- If the tasks are imperfectly aligned ($0 < \rho < 1$), then $\tau_0^* > \tau_S^*$ (transfer requires stronger regularization).

- If the tasks are super-aligned ($\rho > 1$), then $\tau_0^* < \tau_S^*$ (transfer requires weaker regularization).

This phase transition depends solely on task alignment and holds for all noise levels $\sigma_0 > 0$.

*Proof.* Recall from the proof of Theorem 3.8 that the optimal shrinkage factors $x^*$ (transfer) and $x_S^*$ (source) satisfy:

$$G(x^*) = \langle w_0, w_1 \rangle \quad \text{and} \quad G(x_S^*) = ||w_0||^2,$$

where $G(x)$ is a strictly increasing function. Recall also that $x(\tau_0) = 1 - \tau_0 a_0$ is a strictly decreasing function of the regularization $\tau_0$.

- Case 1: Imperfect alignment ($\langle w_0, w_1 \rangle < ||w_0||^2$). Then $G(x^*) < G(x_S^*)$, which implies $x^* < x_S^*$. Since $x$ decreases with $\tau_0$, this implies $\tau_0^* > \tau_S^*$.

- Case 2: Super-alignment ($\langle w_0, w_1 \rangle > ||w_0||^2$). Then $G(x^*) > G(x_S^*)$, which implies $x^* > x_S^*$. Since $x$ decreases with $\tau_0$, this implies $\tau_0^* < \tau_S^*$.

$\square$

