# OpenReview forum: "Source-Optimal Training is Transfer-Suboptimal"
_TMLR — Accepted by TMLR_

### Review · Reviewer_bP5v · 2026-01-30

**Summary Of Contributions:**

The submission presents a series of theoretical results that demonstrate that, in the L2-distance to Starting Point (L2-SP) setting for linear regression (i.e., regularized by $\lambda\|w - w_0\|_2^2$, were $w_0$ is a set of pretrained weights), the regularization penalty that optimizes source/pretraining performance differs from the regularization penalty that optimizes target/downstream performance. In particular, if the optimal weights for source and target are imperfectly aligned (i.e., $w_0^\top w_1/\|w_0\| < 1$, the usual case), then the optimal regularization for the target domain is stronger than the optimal regularization for the source domain. The authors validate their theoretical findings with simulated results in the linear setting and then extend them to 3 experiments using neural nets. In all cases, the optimal regularization penalty for target performance differed substantially from the optimal penalty for source performance, as predicted by the theory.

######## Strengths ########

- With the exception of some passages early in the paper (details below), the manuscript is very well written and easy to follow (despite its complexity). Results and their significance are all carefully explained, with meaningful connections to standard practice beyond the limiting assumptions of the theory.
- The setting studied in the paper, how source training should be modified in order to optimize target performance, has received little-to-no attention in the theory community (per the authors' claims and to the best of my knowledge). Finding definitive answers to these questions would have significant impacts to one of today's most commonplace uses of ML: pretraining for downstream performance.
- The empirical evaluation of Section 4 is well designed and the results obtained in Figs. 1 and 2 validate the theoretical findings, even beyond the theoretical settings studied in Section 3.

######## Weaknesses ########

- Early in the manuscript, the authors make heavy use of notation that is only defined multiple pages later. This makes it difficult to follow some of the initial passages in the paper.
    - Abstract
        - $\tau$ and $\rho$ are used in the abstract and first intro paragraph without a definition (not even an informal one)
        - L2-SP setting is also not defined until much later in the introduction
    - Sec 1.1
        - Presumably $\rho=1$ is the measure-zero set? It might be worth stating that explicitly.
        - "The source estimator $\hat{\beta}_0$ naturally targets the source vector $w_0$. However, the optimal initialization for the target task is the projection of the target singal $w_1$ onto the source, which corresponds to $\rho w_0$." --> This sentence is impossible to decipher with the information given up to this point. What is the "source vector $w_0$" and the $target signal w_1$? What is the alignment $\rho$? When the authors discuss "optimal initialization", does it imply that the transfer mechanism is finetuning? Or do the authors use "initialization" to refer to the value $a$ used in L2-SP for the regularization term $\|\beta-a\|$? (It seems to be either/both, but without an explanation it's just confusing.)
            - After reading through the rest of the paper, most of these points become clear. I encourage the authors to make it possible for the reader to understand this paragraph before reading the rest of the paper.
            - One remaining concern after reading through the manuscript is that I did not find later a statement or proof that the optimal initialization is the projection $\rho w_0$. Could the authors clarify where this is stated?
        - Manuscript again refers to "transfer optimal regularization $\tau_0^*$" without defining it. Presumably $\tau$ is the regularization scale hyperparameter? (The point is that if the authors will not define the notation until later, then it is not useful to the reader to encounter the notation. It is indeed confusing.)
    - Sec 1.2
        - $n_1$, $\sigma_1$ toward the end are undefined.
        - "DE characterization" is not defined — what does DE stand for?
        - Fifth paragraph gives the first definition of $\tau_0^*$ after it has been used multiple times.
- The manuscript provides meaningful connection to the pretrain -> finetune paradigm that governs much of ML today. However, two key points remain insufficiently unaddressed: (1) how the argument that finetuning implicitly regularizes toward initialization applies to networks with very flat local minima, and 2) how applicable these findings may be to a broader setting with multiple unknown target tasks.

**Additional Comments:**

The following points are provided as feedback to hopefully help better shape the submitted manuscript, but will not impact my recommendation in a major way.

Sec 2
- $n_i$ and $p$ should have been defined in Sec 1.2 (they are not actually defined in 1.2 or this section).
- What is the significance of the "S" superscript in $\hat{\beta}_1^\text{S}$? Is it "S" for "standard"? Easily conflated with "source". Maybe best to just use $\hat{\beta}_1$? ("T" would also easily be confused for "transfer" instead of "target")
- Why switch from $\beta$ to $\theta$ to define trained parameters when discussing SGD?

Sec 3.1
- The phrasing "starting from zero versus starting from the source parameters" still seems awkward, since there isn't necessarily any initialization or starting point, but more so an anchor for the regularization.
- Theorem 3.3 needs more explanation. What does it mean that the threshold is determined by "source bias and noise"? All RHS terms include $M^{(1)}$ and $\Sigma_1$. How do these terms affect the expectations on the RHS? Clearly they disappear in the case considered in Corollary 3.4, which does trivially match the authors' description of source bias and  noise.
- One comment that applies to all theoretical results (lemmas, proofs, corollaries): As this is a theory paper, the theoretical results are the key findings, and their derivation is the crux of the methodology. In consequence, it is important for the main manuscript to 1) provide a proof sketch or summary of the key steps in the proof and 2) refer the reader to the relevant appendix where the proof is provided. For (2), it may be sufficient to write at the start of Sec 3 that all detailed proofs are in the appendix, but 1) should be included for all results.
- The remark after Corollary 3.4 is interesting and helpful.

Sec 3.2
- Theorem 3.8 "fixed source model overparameterization $\gamma_0$" — I believe this is the first time in Sec. 3 that the authors mention overparameterization. Presumably the assumption is $\gamma_0 > 1$?
- Is it possible to arrive at a theorem/corollary similar to 3.8/3.9 for the non-isotropic setting? (I'm not asking the authors to do so, just to outline whether the mathematical tools used here could be used for that purpose, or why that is not possible.)

Sec 4
- Please clarify if the data is isotropic
- Notation consistency:
    - $n_{source}$ --> $n_0$
    - $n_{target}$ --> $n_1$.
    - $\gamma$ --> $\gamma_0$
    - $\lambda_S$ --> $\tau^*_\text{S}$
    - $\lambda_{TL}$ --> $\tau^*_0$
- "logarithmic grid" --> "logarithmic sequence" or "logarithmically spaced range" ('grid' seems awkward for 1D).
- Fig. 1 is a really nice result.
    - Just like Fig. 2 goes beyond the assumptions in the theory, it may be meaningful to provide results somewhere in-between. For example, if Fig. 1 considers the isotropic setting, how would these results look in the non-isotropic setting? How do Fig. 1 results vary in the face of varied over-parameterization values $\gamma$? These may enable extracting insights about how broadly the theory could be expanded, even within the linear setting.
- Fig. 2 is also really nice.
- In the NN case, are all model layers (including the output head) initialized with source model weights, or is any part of the model reinitialized at random?

Sec 5.2
- "The recent work ... on gradient-descent-based fine-tuning provides complementary insights" — could the authors expand on what these insights are and how they are complementary?

Typos/style/grammar
- Sec 3.1 "noteable" -> "notable"

**Audience:**

Yes

**Audience Explanation:**

The specific findings of this work are likely significant for a niche audience focused on transfer learning theory. However, the authors do a good job of connecting the findings to the more broadly relevant paradigm of pretraining and finetuning. In this front, I think the authors could improve their manuscript by:
1. Discussing the setting of flat local minima. The connection to standard finetuning in Sec. 2 is meaningful and intuitively reasonable. But I do not believe that it holds for networks with flat minima, where the basin of the initialization is large. Since modern neural net architectures seek to create such flat optima (e.g., see [1]), it may be worth discussing the effect of such wider basins.
2. Discussing the multiple-target setting in more depth. The setting considered in the theoretical results assumes one source task and one (known) target task. This is interesting and a good starting point, but I'm curious what it would take to extend these findings to 1 source task N target tasks, and how we could leverage these insights if the N target tasks are unknown—this setting matches the most common use case today of representation learning. This point is brought up as future work in Sec. 5.2, but a bit of insight into what the challenges are or how big a leap it would be to go from the current results to the multi-target setting would be meaningful.
    - One practical implication that would be useful to spell out is how to choose the source regularization. For example, if one has access to downstream task data during source training, one could use validation error on the target task to choose the source regularization penalty. More typically, source training is expensive and often happens only once, without access to the exact target task(s). How should one choose the source regularization in such settings?

[1] Hao Li et al. "Visualizing the loss landscape of neural nets." NeurIPS-18.

**Claims And Evidence:**

Yes

**Claims Explanation:**

The manuscript does an excellent job of laying out the key contributions up-front in the introduction, where each result and its significance is briefly described. These results are later validated theoretically, and the main result (the divergence between source-optimal and target-optimal regularization penalty) is also empirically validated.

**Requested Changes:**

No changes are critical to my recommendation. The weaknesses and comments under the "Interest" box outline some of the main points I believe would strengthen the submission. I also provide some suggestions in the "Additional comments" box that would also strengthen the submission, but to a lesser extent.

---

> ### Author Response · Authors · 2026-04-01
> **Resubmission Response**
>
> We thank the reviewer for the thorough and detailed feedback, which has led to substantial improvements. We particularly appreciate the detailed section-by-section feedback, which has improved the exposition throughout. We respond to each point below.
>
>
> Weakness 1. Early notation undefined.
>
> We have overhauled the abstract and introduction: L2-SP is defined parenthetically, τ_S* and τ_0* are described at first use, ρ is defined explicitly in Section 1.1, "optimal initialization" is replaced with "optimal anchor for the L2-SP penalty," DEs are introduced by name, and the "S" superscript is clarified as "standard." Regarding ρw_0 being the optimal anchor: this follows from the proof of Theorem 3.8, where the transfer risk is minimized at the shrinkage factor satisfying G(x*) = ρ‖w_0‖². We have added a proof sketch in Section 3.2 making this explicit.
>
>
> Weakness 2. Flat minima and multi-target settings.
>
> On flat minima: we acknowledge in Section 5.2 that SGD may travel far from initialization in flat landscapes, weakening the L2-SP analogy. The new parameter distance plots (Figure 4) empirically bound the drift in our setting. On multi-target: Section 5.2 now discusses the extension to N targets as a minimax/average-case problem and notes that over-regularization is robust across the imperfect alignment regime even without knowing ρ or the target distribution.
>
>
> Interest box: How to choose source regularization in practice.
>
> The key insight is that over-regularization is optimal across the entire imperfect alignment regime (0 < ρ < 1), so increasing regularization beyond source-optimal is a well-motivated default even without target task information. When downstream data is available, cross-validation on transfer performance can select the source penalty directly.
>
>
> Sec 2: "S" superscript, β vs θ notation.
>
> Clarified. θ appears only when discussing SGD-based fine-tuning; β is used throughout for ridge estimators.
>
>
> Sec 3.1: "Starting from zero" phrasing; Theorem 3.3 explanation; proof sketches.
>
> Revised to "regularizing toward zero versus regularizing toward the source parameters." Expanded the paragraph preceding Theorem 3.3 to explain the RHS terms. Added proof sketches for every result in Section 3.
>
>
> Sec 3.2: γ_0 > 1 assumption; non-isotropic extension.
>
> γ_0 > 1 is now stated explicitly. A new Remark after Corollary 3.9 discusses the non-isotropic case and why it depends on alignment relative to the eigenbasis of Σ_0.
>
>
> Sec 4: Isotropic clarification; notation; logarithmic grid; varied γ_0 and non-isotropic results; NN initialization.
>
> All addressed. Synthetic data explicitly states Σ_0 = Σ_1 = I. Figure 2 now has subfigures for the γ_0 sweep and power-law covariance sweep. Transfer protocol specifies that all layers are initialized from source and the classification head is reinitialized.
>
>
> Sec 5.2: Ghane et al. complementary insights.
>
> Expanded: their results characterize the iterative GD regime while ours address the closed-form L2-SP solution; together these bracket practical fine-tuning.
>
>
> Typo: "noteable."
>
> Fixed.

---

> > ### Comment · Reviewer_bP5v · 2026-04-06
> > **All concerns addressed**
> >
> > All my concerns were addressed by the authors' responses.

---

### Review · Reviewer_a3RW · 2026-03-08

**Summary Of Contributions:**

This paper studies the source-side optimization problem in $L_2$-SP ridge regression for transfer learning. It proves that the source regularization optimal for the source task generically differs from the one optimal for downstream transfer ($\tau_0^* \neq \tau_S^*$ outside a measure-zero set). An alignment-dependent phase transition is identified: imperfect alignment ($\rho < 1$) favors stronger source regularization, while super-alignment ($\rho > 1$) favors weaker. In isotropic settings, whether transfer helps is independent of target sample size and noise. Synthetic ridge experiments validate the theory; nonlinear experiments on MNIST, CIFAR-10, and 20 Newsgroups suggest the qualitative pattern persists.

**Additional Comments:**

- Theorem 3.8 asserts existence and uniqueness of $\tau_0^*$ but no closed-form is given, even in the isotropic case. Providing one would improve interpretability.
- Several notational inconsistencies in the appendix ($\delta$ vs. $\tilde{\delta}$, trace expressions) make end-to-end verification difficult.
- The "measure-zero" separation claim could be stated more precisely; the set where $\tau_0^* = \tau_S^*$ is simply $\{\rho = 1\}$, but this deserves explicit mention.

**Audience:**

Yes

**Audience Explanation:**

See Summary Of Contributions

**Claims And Evidence:**

Yes

**Claims Explanation:**

1. The paper poses a clean, underexplored question — optimizing the source model for transfer rather than for its own task — and delivers a logically tight answer in the linear setting. The chain from finite-sample formulas to deterministic equivalents to the phase transition is well-structured and the proofs are self-contained.

2. The independence of the transfer decision from target-side quantities ($n_1$, $\sigma_1$, $\tau_1$) in the isotropic regime is a sharp and surprising result. The geometric intuition — source estimator scale vs. target projection scale $\rho w_0$ — provides a compelling mental model.

3. The synthetic experiment faithfully validates the predicted phase transition at $\rho = 1$, including quantitative predictions (e.g., $\sim 1.6\times$ ratio at $\rho = 0.8$). This is a strong sanity check for the theory.

**Requested Changes:**

1. The gap between theory and nonlinear experiments is the paper's most significant limitation. The nonlinear experiments use standard SGD fine-tuning with weight decay, not $L_2$-SP. The appeal to "implicit regularization" is never formalized or empirically verified (e.g., by measuring $\|\theta - \theta_0\|$ during training). More critically, the effective alignment $\rho$ is never estimated in the nonlinear setting, so one cannot distinguish the paper's specific mechanism from the generic observation that heavier regularization produces more transferable features.

2. The theoretical setting (isotropic covariance, known $\rho$, $L_2$-SP ridge) is quite restrictive, and the technical machinery (Bai–Silverstein deterministic equivalents, Silverstein fixed-point equations) is standard. The marginal contribution over prior work [1][2][3] is a valid but narrow shift from "optimize target given fixed source" to "optimize source given fixed target protocol." The non-isotropic case (Theorem 3.6) is stated but never concretely instantiated or experimentally probed.

3. The nonlinear experimental design is weak by current standards. Architectures are small and dated (2-layer MLP, 2-conv CNN). Task splits are ad hoc (round vs. angular digits) without alignment-motivated construction. The weight decay grid is coarse (8 values), only 5 seeds are used, and no ablations vary $\sigma_0$ or $n_1$ to test key theoretical predictions (monotonic shrinkage of transfer region with $\sigma_0$; target-side independence). Including an explicit $L_2$-SP fine-tuning baseline would directly test the theory's relevance.

4. The super-aligned regime ($\rho > 1$) is never demonstrated in nonlinear experiments, and the authors acknowledge it may be "difficult to realize." Since this is one of the two halves of the claimed phase transition, its absence significantly weakens the generality claim. Additionally, "super-alignment" under the paper's definition ($\rho = \langle w_0, w_1 \rangle / \|w_0\|^2 > 1$) is non-standard relative to cosine similarity conventions; more discussion of when this realistically arises would help.

5. The practical framing ("implications for foundation model training") overstates what the results support. The recommendation to "use more regularization for transfer" is already common practice and was noted in the original $L_2$-SP paper [4]. The assumption that $\rho$ is known or estimable at pretraining time is unrealistic. The paper would also benefit from engaging with recent empirical work on pretraining weight decay and downstream plasticity, and from comparing to related linear transfer estimators.
- [1] "Double Double Descent: On Generalization Errors in Transfer Learning between Linear Regression Tasks."
- [2] "The Common Intuition to Transfer Learning Can Win or Lose: Case Studies for Linear Regression."
- [3] "Universality in Transfer Learning for Linear Models."
- [4] "Explicit Inductive Bias for Transfer Learning with Convolutional Networks."

---

> ### Author Response · Authors · 2026-04-01
> **Resubmission Response**
>
> We thank the reviewer for the thorough and detailed feedback, which has led to substantial improvements. Below we respond to each concern.
>
>
> Q1. Theory-practice gap: no L2-SP baseline, no ‖θ − θ_0‖ measurement, no alignment estimation in the nonlinear setting.
>
> We agree this was a significant limitation. We now run an explicit L2-SP fine-tuning baseline alongside standard SGD across all three experiments (Figure 3), track parameter distance from initialization (Figure 4), and add a target-side independence ablation (Figure 5) sweeping target fractions from 1% to 50%. The target-independence test is a specific, falsifiable prediction of our mechanism; a generic "heavier regularization = more transferable features" explanation would not predict that the optimal source regularization is constant across a 50× range of target data. Regarding alignment estimation, we have added a discussion of proxy measures in Section 5.2, though we believe the target-independence test is a more direct test of our mechanism.
>
>
> Q2. Restrictive theoretical setting; non-isotropic case never probed.
>
> We have added a γ_0 sweep (Figure 2a) and a non-isotropic covariance experiment with power-law spectra (Figure 2b), plus a Remark after Corollary 3.9 discussing the non-isotropic extension.
>
>
> Q3. Experimental design critiques.
>
> Weight decay grid expanded from 8 to 19 values, seeds from 5 to 10, and we have added the L2-SP baseline, parameter tracking, and target-independence ablation described above. We use smaller architectures deliberately so that source-side regularization effects are clean and reproducible; the consistent pattern across MLP, CNN, and TF-IDF+MLP suggests the phenomenon is not architecture-specific.
>
>
> Q4. Super-aligned regime absent in nonlinear experiments.
>
> We have added a dedicated paragraph in Section 5.2 discussing why super-alignment is structurally unlikely in deep networks (feature co-adaptation, disjoint label splits). We note that ρ > 1 means the target amplifies the source-relevant direction; it is not a cosine similarity but captures the scale mismatch that drives the regularization trade-off.
>
>
> Q5. Practical framing overstated.
>
> We have toned down the foundation model claims and clarified our contribution relative to the original L2-SP paper: the observation that over-regularization helps is not new, but we make it analytically precise, characterize the divergence as a function of alignment, and show it is independent of target-side quantities.
>
>
> Additional: No closed-form for τ_0*.
>
> The proof sketch for Theorem 3.8 (Section 3.2) now provides the implicit characterization via G(x*) = ρ‖w_0‖².
>
>
> Additional: Measure-zero claim imprecise.
>
> The proof now explicitly identifies the set as {ρ = 1}.
>
>
> Additional: Notational inconsistencies in appendix.
>
> Reviewed and corrected.

---

### Review · Reviewer_EsMf · 2026-03-16

**Summary Of Contributions:**

The authors prove that optimally training a source model is generally suboptimal for transferred task. They then model the transfer-optimal regularization as a function of task alignment, formulating phase transition - transfer usually prefers stronger source regularization when tasks are imperfectly aligned, and weaker source regularization when tasks are super-aligned.

**Audience:**

Yes

**Audience Explanation:**

The topic of fine-tuning has a large amount of audiences as its recent trends, who would gain insight from this work and could try different regularizations in source and target in doing fine-tuning based on the claim of this paper.

**Broader Impact Concerns:**

N/A.

**Claims And Evidence:**

Yes

**Claims Explanation:**

Claims are well supported and evidenced in both theories and numerical simulations on ridge regression and L2-SP transfer learning, and some practical evidence in ablating weight decay of shallow neural net training.

**Requested Changes:**

Some questions:
1. The alignment $\rho$, does it really reflect the alignment between the two task? Geometrically it's the projection of $w_1$ on $w_0$'s direction. They could form a big angle and evaluate to a big score if the vectors are not normalized. And realistically, does it make good sense to evaluate alignment of two tasks by distance of two model parameters? Would it make more sense to measure the distance between the two distributions or "underlying ground-truth model functions"? Another concern is, for practical transfer learning, it's very difficult to measure the alignment of the source and target tasks - in other words, when a researcher wants to use claim of this work to do fine-tuning, how would a researcher decide how to do source and target regularization?
2. The settings of the theoretical claims are in linear and ridge regression settings - would be nice and important to have a proxy that connects this setting to a nonlinear setting because practical models for fine-tuning are mostly not linear models, on which a strong theory could be a big advantage for the claim of this paper.
3. The different setting from theory to real-world data experiment: Theories are on ridge regression and L2-SP transfer learning, but those experiments in section 4.2 use shallow MLP/CNN, although weight decay is very similar to ridge regularization parameter. Would be nice to have some theoretical bridge to fill this gap. Also, the training methods is not clear - does it use SGD, what's the optimizer, what's the fixed fine-tuning protocol? They are crucial for justifying the claim.
4. Nit: Notations are sometimes confusing - $\lambda_0$ and $\lambda_1$ denote regularization parameter of source and target training, while $\tau^{\ast}\_{0}$ and $\tau^{\ast}\_{S}$ denote transfer- and source-optimal regularization.

---

> ### Author Response · Authors · 2026-04-01
> **Resubmission Response**
>
> Response to Reviewer EsMf
>
> We thank the reviewer for their careful reading and constructive feedback. We have revised the manuscript to address all points raised; below we respond to each.
>
>
> Q1. The alignment ρ, does it really reflect the alignment between the two tasks?
>
> The quantity ρ = ⟨w_0, w_1⟩ / ‖w_0‖² is asymmetric by design; it measures how much of the source signal is useful for the target, which is the natural quantity for source-side optimization. The asymmetry captures the scaling mismatch between what the source estimator produces and what the target needs. We agree that ρ is not directly computable for deep networks. In the revision we have added a paragraph on practical alignment estimation (Section 5.2) discussing proxy measures such as CKA and task2vec. Importantly, the theory does not require knowing ρ at pretraining time; it explains why over-regularization is robust across the entire imperfect alignment regime (0 < ρ < 1), so the practical recommendation does not require estimating ρ precisely.
>
>
> Q2. Would be nice to have a proxy that connects the linear setting to a nonlinear setting.
>
> We have added an explicit L2-SP fine-tuning baseline (Figure 3), parameter distance tracking (Figure 4), and a target-side independence ablation (Figure 5) to bridge this gap. Together these provide empirical evidence that the L2-SP mechanism is operative in standard nonlinear fine-tuning.
>
>
> Q3. Training methods are not clear — optimizer, fine-tuning protocol?
>
> Section 4.2.1 now specifies all details: Adam optimizer, learning rate, batch size, head reinitialization, L2-SP penalty, and 10 seeds.
>
>
> Q4. Notation note.
>
> We have defined all notation at first use in the abstract and introduction, including parenthetical definitions of τ_S* and τ_0*.

---

### Author Response · Authors · 2026-03-31
**Brief Comment to Reviewers**

We thank all three reviewers for their careful and constructive feedback. We are currently preparing a revised manuscript and wanted to briefly outline the changes underway.

The most consistent concern across reviews was the gap between our L2-SP ridge theory and the nonlinear experiments, which use standard SGD fine-tuning with weight decay rather than explicit L2-SP. We agree this was the paper's most significant limitation. In the revision we are adding an explicit L2-SP fine-tuning baseline across all three nonlinear experiments, directly testing the theoretical mechanism in neural networks. We are also tracking parameter distance from initialization during fine-tuning to empirically verify the implicit L2-SP claim.

On the experimental side, we have substantially strengthened the design: the weight decay grid is expanded from 8 to 19 values, seeds are increased from 5 to 10, and we are adding a target-side independence ablation that sweeps target data fraction to test the prediction that the transfer-optimal source regularization ranking is preserved across target sample sizes. For the synthetic experiments, we are adding a sweep over overparameterization levels and a non-isotropic covariance experiment with power-law spectra, probing Theorem 3.6 beyond the isotropic setting.

We are also overhauling the notation in the introduction per Reviewer bP5v's detailed feedback, adding proof sketches for the main results, providing the implicit characterization of $\tau_0^*$ in the isotropic case, and expanding the discussion to address flat minima, multi-target settings, practical alignment estimation, and the absence of the super-aligned regime in nonlinear networks. We are softening the practical claims in Section 5 to better reflect the scope of our results.

These experiments are currently running and we expect to submit the revised draft shortly. We believe the changes address all major concerns raised by the reviewers and meaningfully strengthen both the theoretical presentation and empirical validation.

Thank you all again for your reviews.

---

### Author Response · Authors · 2026-05-03
**Github repo**

Please see https://github.com/evanshedges2/transfer_optimal_l2sp for experimental code used in preparation of this manuscript.

---

### Decision · Action_Editor_EjRx · 2026-04-30

**Recommendation:** Accept as is

**Audience:**

Yes

**Audience Explanation:**

Yes. Regularisation, transfer-learning and domain adaptation are all topics that are interesting to members of TMLR's audience. The findings of the paper are known experimentally by many, and it is a solid contribution to the community to provide some theoretical support for these observations.

**Claims And Evidence:**

Yes

**Claims Explanation:**

The paper presents a series of theoretical results showing that when doing linear regression regularised by the L2-distance to the initialisation, the regularisation penalty that optimises the source dataset performance differs from the penalty that optimises downstream dataset performance. These findings are then validated experimentally.

Reviewers all agreed that both the theoretical and experimental results are compelling and well-substantiated.